# Probing dark exciton navigation through a local strain landscape in a WSe$_2$ monolayer

Ryan J. Gelly[1,7], Dylan Renaud[2,7], Xing Liao[3], Benjamin Pingault[2], Stefan Bogdanovic[2], Giovanni Scuri [1], Kenji Watanabe [4], Takashi Taniguchi [5], Bernhard Urbaszek [6], Hongkun Park[1,3✉] & Marko Lončar [2✉]

In WSe$_2$ monolayers, strain has been used to control the energy of excitons, induce funneling, and realize single-photon sources. Here, we developed a technique for probing the dynamics of free excitons in nanoscale strain landscapes in such monolayers. A nanosculpted tapered optical fiber is used to simultaneously generate strain and probe the near-field optical response of WSe$_2$ monolayers at 5 K. When the monolayer is pushed by the fiber, its lowest energy states shift by as much as 390 meV (>20% of the bandgap of a WSe$_2$ monolayer). Polarization and lifetime measurements of these red-shifting peaks indicate they originate from dark excitons. We conclude free dark excitons are funneled to high-strain regions during their long lifetime and are the principal participants in drift and diffusion at cryogenic temperatures. This insight supports proposals on the origin of single-photon sources in WSe$_2$ and demonstrates a route towards exciton traps for exciton condensation.

[1] Department of Physics, Harvard University, Cambridge, MA 02138, USA. [2] John A. Paulson School of Engineering and Applied Sciences, Harvard University, Cambridge, MA 02138, USA. [3] Department of Chemistry and Chemical Biology, Harvard University, Cambridge, MA 02138, USA. [4] Research Center for Functional Materials, National Institute for Materials Science, 1-1 Namiki, Tsukuba 305-0044, Japan. [5] International Center for Materials Nanoarchitectonics, National Institute for Materials Science, 1-1 Namiki, Tsukuba 305-0044, Japan. [6] Université de Toulouse, INSA-CNRS-UPS, LPCNO, 135 Avenue Rangueil, 31077 Toulouse, France. [7] These authors contributed equally: R.J. Gelly, D. Renaud. ✉email: hongkun_park@harvard.edu; loncar@seas.harvard.edu

WSe$_2$ monolayers are direct bandgap semiconductors whose optical properties are dominated by the presence of bound electron-hole pairs, namely excitons. WSe$_2$, a member of the larger class of semiconducting transition metal dichalcogenides (TMDs), has several attractive properties: its monolayers have an optically dark-exciton ground state[1-3], can host single-photon sources[4,5], and have potential for efficient exciton funneling[6-8]. Critically, the single-photon sources and exciton funneling schemes rely on the presence of local strain landscapes that are imposed on the WSe$_2$ monolayer.

Previous studies of local strain effects in atomically thin semiconductors have been limited to static[4,5,9-12], or room-temperature[7,13,14] situations. Important advancements towards dynamically applied strains in a cryogenic environment have so far remained in the low strain regime (<0.5%)[6]. Consequently, the precise effects of local strain on excitonic properties at high strain could not be studied in detail. Here, we systematically address the role of locally applied strain on free (as opposed to defect-bound) excitonic species in WSe$_2$ monolayers by dynamically generating sub-micron scale (~100 nm) strain at cryogenic temperatures. We use the tip of a nanosculpted tapered optical fiber to locally deform the monolayer and to optically probe the region (Fig. 1a and Supplementary Fig. 1). By mounting the optical fiber on a piezoelectric nanopositioner, we are able to controllably and reversibly strain the suspended WSe$_2$ monolayer. Optical characterization is achieved by using the fiber's fundamental mode to both excite and collect emission at the fiber's facet. We engineer the fiber to have a tip radius of 240 nm to maximize the coupling efficiency of ~700 nm light while still retaining well-localized excitation and collection profiles (Fig. 1b and Supplementary Fig. 2). We encapsulate a WSe$_2$ monolayer between two layers of hexagonal boron nitride (hBN) in order to increase the resistance to tearing[15]. Encapsulation and suspension of WSe$_2$ monolayers also leads to improved, spatially homogeneous optical properties[16,17].

## Results

### Strain-dependent spectroscopy via a tapered optical fiber.
Figure 1c shows the fiber-collected transmittance spectrum of a proximal, strain-free hBN/WSe$_2$/hBN heterostructure **D1** in a cryogenic environment ($T = 5$ K), measured by illuminating it with white light from the backside. The spectrum shows four transmittance dips that correspond to the A and B excitons (X$_{A:1s}$ and X$_{B:1s}$)[18,19] and their first excited Rydberg counterparts (X$_{A:2s}$ and X$_{B:2s}$)[18,19]. Figure 1d presents a fiber-collected PL spectrum from **D1** measured by exciting through the fiber, again without an applied strain at 5 K. The PL spectrum shows a different set of four prominent peaks, originating from (in descending order in energy) the spin-allowed, neutral, bright exciton (X$^0$)[18], charged exciton (X$^-$)[20], nominally spin-forbidden, z-polarized, dark exciton (D$^0$)[1-3], and a phonon replica of the dark exciton (D$^R$)[21,22]. We detect not only bright excitons whose transition dipole moment lies in the WSe$_2$ plane, but also dark excitons whose transition dipole moments lie out-of-plane[3] because the fiber is in close proximity with **D1** and couples to the near-field (Fig. 1b and Supplementary Fig. 2). Therefore, the strong intensity of free dark excitons in Fig. 1d is a direct consequence of using a tapered optical fiber for interrogating the WSe$_2$ monolayer.

We now apply a controlled strain to an hBN/WSe$_2$/hBN heterostructure by pushing the fiber tip against it with a piezoelectric nanopositioner. We simultaneously monitor both transmission and PL spectra as a function of the voltage applied to the piezo-positioner ($V_p$). For device **D2**, when we apply $V_p < 1$ V, the dips in the transmission and the peaks in the PL

spectra red-shift with increasing $V_p$ (Fig. 1e, f, respectively). We also observe some broadening in the X$_{A:1s}$ state that may be due to inhomogeneous broadening stemming from some non-uniformity in the strain imposed by the fiber facet. The broadening may also originate from non-radiative broadening due to the coupling of different valleys by strain. Tracking the Rydberg states of the A exciton as a function of strain reveals only a small (~2 meV) change in the exciton binding energy (Supplementary Fig. 3). Thus, we attribute shifts in PL primarily to the strain-induced change in bandgap[23,24].

### PL and transmittance spectra at very high strains.
As we continue pushing the hBN/WSe$_2$/hBN heterostructure by applying larger $V_p$, its spectral response changes dramatically. The transmittance spectra, measured in device **D3**, plateaus as $V_p$ increases to 10 V (Fig. 2a). The PL, on the other hand, exhibits a radically different spectrum (Fig. 2b) at $V_p = 10$ V, compared to $V_p = 0$. As shown in Fig. 2c, the PL spectrum branches abruptly into two distinct sets at around $V_p \sim 1.5$ V. The higher energy (~1.7 eV) set, consisting of a number of plateauing features, does not exhibit an energy shift in the $V_p$ range of 1.5–10.0 V, similar to the transmittance spectra. The lower energy (<1.6 eV) red-shifting set, on the other hand, shifts in energy by more than 390 meV in the same $V_p$ range, reaching 1.38 eV at $V_p = 10.0$ V. Remarkably, this energy shift corresponds to over 20% of the bandgap of an unstrained WSe$_2$ monolayer.

### Dark excitons unveiled by polarization- and time-resolved PL.
Insight into the origin of the two branches is provided by the polarization-selective PL spectra from **D3** in Fig. 3a–c that are confocally measured from the side of the heterostructure. The lower energy (<1.6 eV), red-shifting PL peaks that appear in the high-$V_p$ (and thus high-strain) regime are predominantly polarized out of the plane of the WSe$_2$ monolayer, while the higher energy (1.7 eV), plateauing PL peaks are not z-polarized. This observation indicates that the lower energy red-shifting PL peaks stem from the out-of-plane polarized dark excitons, while the higher energy PL peaks are primarily from bright excitons[1,3]. Charge tuning measurements provide additional evidence that free dark excitons and their phonon replicas feature prominently in the red-shifted peaks on the basis of known energetic splittings and intrinsic regime widths[22,25] (Supplementary Fig. 4).

The data in Figs. 2 and 3 demonstrate that both bright and dark excitons play important roles in determining the optoelectronic response of a WSe$_2$ monolayer in the presence of strain. To better understand the nature of these roles, we utilize time-resolved PL to directly probe the exciton dynamics. Using a 1 ps above-bandgap pulsed laser (400 nm) and a streak camera (2 ps resolution), we probe the integrated intensity from the red-shifted features and the plateauing features in device **D4** (Fig. 4a, associated spectrum in Supplementary Fig. 5). The associated lifetimes are 5 ps for the plateauing features and 48 ps for the red-shifted features. The longevity of the red-shifted PL peaks compared to the plateauing counterpart is consistent with the identification of the two sets as dark and bright excitons, respectively[26]. Moreover, because the lifetimes are on the picosecond scale, we know we are looking at free excitons, as opposed to single-photon emitter states with lifetimes greater than 1 ns. Notably, in addition to the differing lifetimes, the two sets have different early time behaviors as well. The bright exciton population rises concurrently with the laser pulse, while the dark-exciton population does not reach its peak until after a ~10 ps delay. This clearly resolvable rise time delay indicates that dark excitons in the red-shifted branch populate more slowly than excitons in the high-energy branch.

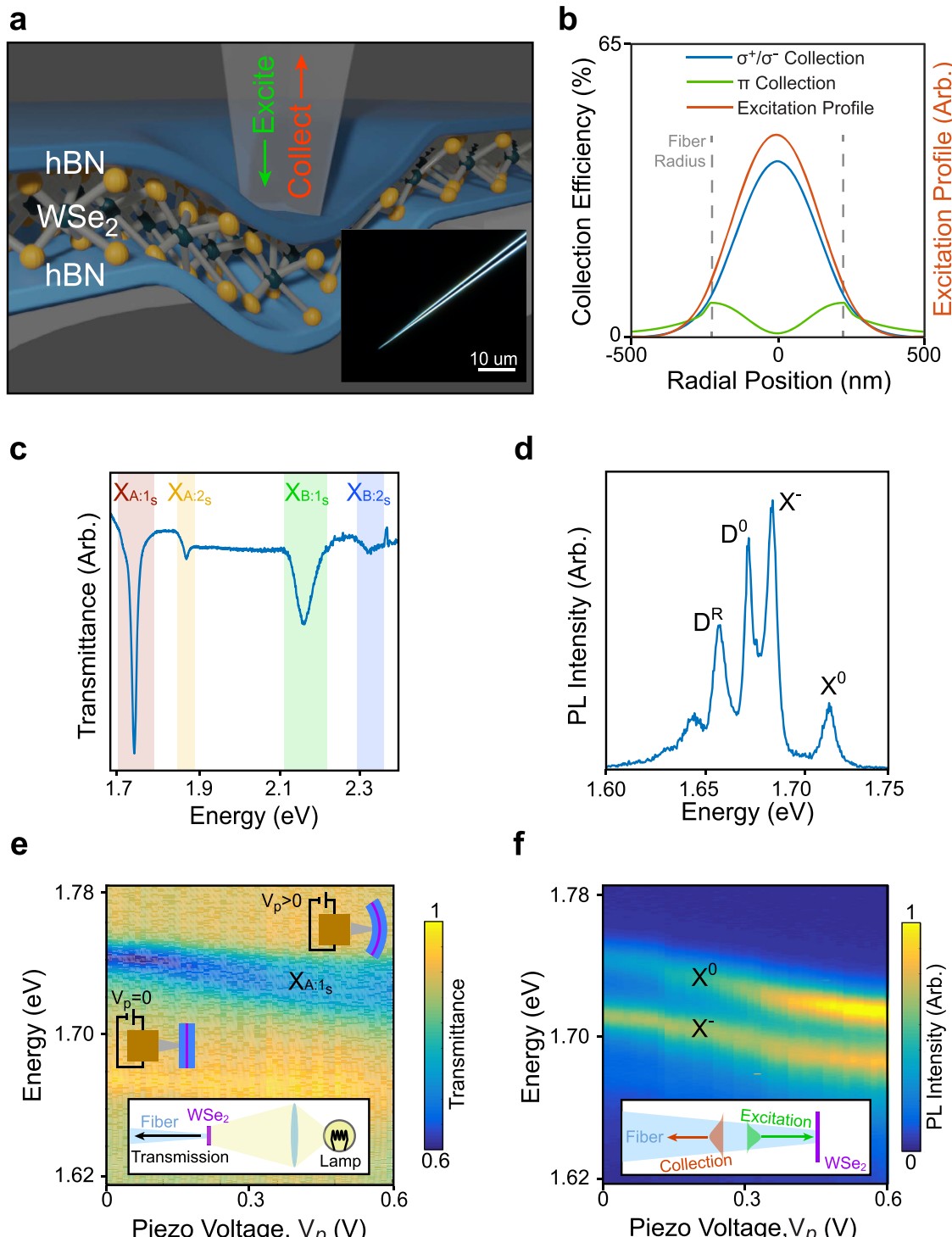

**Fig. 1 A cryogenic, fiber-based technique for strain-dependent spectroscopy. a** A schematic of the fiber interfacing with an hBN/WSe₂/hBN heterostructure. The fiber is mounted on a piezoelectric nanopositioner (not pictured). Inset: Darkfield optical micrograph of a 240 nm radius tapered fiber. **b** We simulate excitation and collection profiles (for both circularly polarized ($\sigma^+/\sigma^-$) and z-polarized ($\pi$) light, for z axis normal to the WSe₂) for a 240 nm radius fiber tip and 700 nm light. **c** White light transmitted through the device in a cryogenic environment ($T = 5$ K) and collected by the fiber (depicted schematically in inset) shows four transmittance dips. They correspond to the A and B excitons ($X_{A:1s}$, $X_{B:1s}$) and their first excited Rydberg states ($X_{A:2s}$, $X_{B:2s}$). **d** Fiber-collected PL at 5 K shows four pronounced features originating from (in decreasing energy) the neutral exciton ($X^0$), the charged exciton ($X^-$), the dark exciton ($D^0$), and the dark exciton's phonon replica ($D^R$). **e** As we push the device with the fiber by increasing the piezo-positioner voltage ($V_p$), we observe a decrease in the energy of the $X_{A:1s}$ transmittance dip with increased fiber displacement. **f** Likewise, for the same fiber displacement as in **e**, we observe decreases in the energy of the $X^0$ and $X^-$ in PL. Data in **c**, **d** come from device **D1**, while data in **e**, **f** come from **D2**. **D1** and **D2** vary only by few-nm differences in hBN thicknesses.

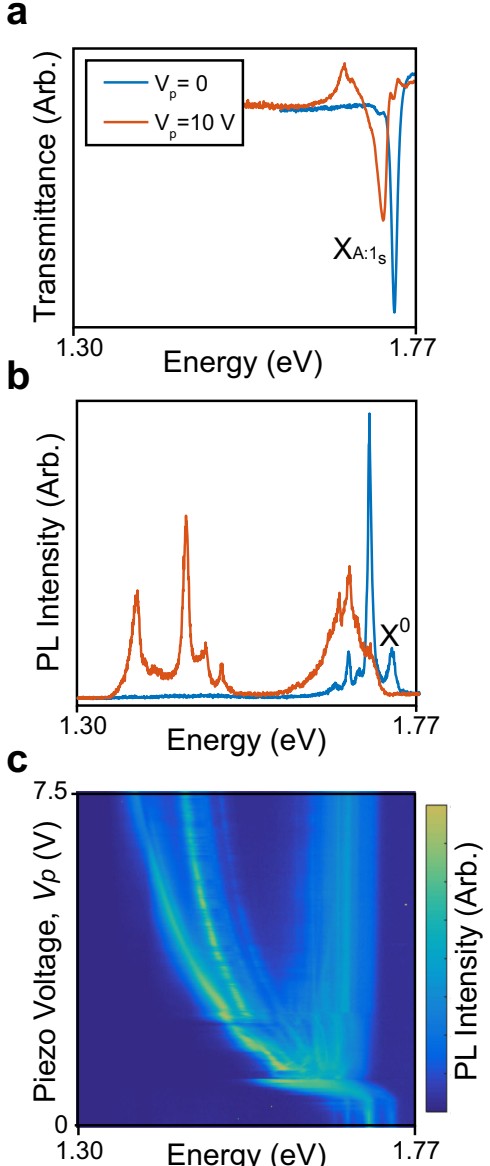

**Fig. 2 Strain-induced excitonic response in transmittance and PL with large tunability at $T = 5$ K. a** For device **D3**, we present the transmittance dip associated with $X_{A:1s}$ with $V_p = 0$ (before contacting the heterostructure with the fiber) and with $V_p = 10$ V (the maximum voltage we can apply to the piezo-positioner). The $X_{A:1s}$ feature shifts by 35 meV over this range. **b** The fiber-collected PL has a feature $X^0$ that matches in energy with $X_{A:1s}$ at $V_p = 0$. At $V_p = 10$ V, peaks as low as 1.38 eV appear. **c** PL spectra as a function of piezo-positioner voltage shows two branches of features. The branch that is lower in energy shifts by 390 meV from the unstrained $X^0$ resonance.

**Numerical modeling of the strain landscape**. An explanation of the observed behavior is provided by considering the spatially-dependent strain profile generated by the fiber. We calculate the strain profile in a hBN/WSe$_2$/hBN heterostructure pushed by a flat fiber using a finite element method (Fig. 4b, see Supplementary Fig. 6). These calculations reveal that there are two regions with qualitatively different strain behaviors. At the fiber facet, the membrane undergoes no appreciable change in strain due to its adhesion to the silica facet[27]. In an annular region at the fiber circumference, however, the tensile strain applied to

the heterostructure reaches 5% when it is displaced 300 nm by the fiber.

This strain landscape from the finite element analysis, combined with the excitation and collection profiles in Fig. 1b suggests a model that explains the excitonic behaviors of a locally strained WSe$_2$ monolayer. In this model, the red-shifting dark-exciton features are associated with the fiber circumference, where the strain increases with increasing displacement. The plateauing features are, on the other hand, associated with excitons at the fiber facet where strain increases minimally. Because the excitation profile is concentrated at the center of the fiber, any excitons that reach the fiber circumference must arrive via diffusion and drift, as depicted in Fig. 4c.

This model explains not only the origin of the two sets of features that we observe at high strain, but also the preponderance of dark excitons in the red-shifted features and the delay in the onset of the time-resolved PL. Drift and diffusion are not instantaneous processes, and the time it takes for the excitons to transit from the excitation site to the circumference explains the 10 ps delay. Other processes, such as the scattering from other states across reciprocal space[28], are too fast to explain this long of a delay. Based on the previously measured diffusivity of dark excitons[29], dark excitons are expected to diffuse 450 nm in 10 ps, roughly equivalent to the length scale in our strain landscape set by the fiber's diameter. Moreover, because this delay time is longer than the bright exciton lifetime, any bright excitons in the system will decay before reaching the circumference. Numerical solutions to the drift-diffusion equation also capture this behavior qualitatively (Supplementary Fig. 7). Dark excitons are the only ones that can reach the circumference, consistent with the out-of-plane polarization of the red-shifted features.

**Lack of exciton transport in strained MoSe$_2$ monolayers**. To see how our interpretation above translates to other material systems, we repeat the strain tuning measurements on an hBN/monolayer MoSe$_2$/hBN heterostructure. At low $V_p$ and thus low strain, we observe a qualitatively similar behavior to a WSe$_2$ monolayer (Supplementary Fig. 8). In the case of MoSe$_2$ heterostructures, however, we do not observe red-shifting features even at the large $V_p$ limit. This observation is consistent with the above interpretation because, unlike the case of a WSe$_2$ monolayer, the dark excitons in a MoSe$_2$ monolayer lie higher in energy than the bright-exciton ground state[30], making dark exciton transport across the strain landscape unobservable.

**Discussion**
The experimental technique that we developed allowed us to locally generate and optically probe strain fields in suspended WSe$_2$ monolayers. Remarkably, using strain, we achieved the red-shift of dark-exciton PL peaks by as much as 390 meV, which corresponds to 20% of the bandgap of an unstrained WSe$_2$ monolayer. We also show that charge control is compatible with straining the device. This exceptional tunability, from the visible to the near-infrared, highlights the potential of two-dimensional materials for realizing novel optoelectronic devices. Finally, we uncovered the role that dark excitons play in the transport of energy across strain landscapes. While it has been understood that these single-photon sources likely originate from the binding of excitons to defects, it has not been clear how the excitons reach the defects efficiently, nor has the essential role of the dark exciton been previously identified experimentally. Therefore, our results support recent theoretical proposals[31] that efficient funneling of free dark excitons is a key ingredient in forming single-photon sources under localized strain. Our results also suggest that the scarcity of strain-induced single-photon sources in

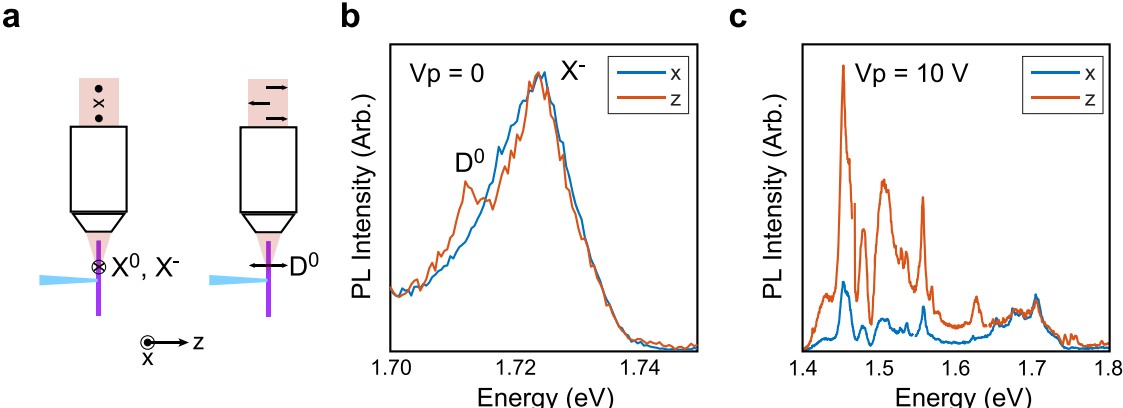

**Fig. 3 Transition dipole moment measurement by polarization-selective PL spectroscopy. a** A scheme whereby polarization-selective, confocal collection of light from the side of the WSe$_2$ monolayer (purple) can distinguish between in-plane (parallel to the device) optical dipoles (such as associated with X$^0$ and X$^-$) and out-of-plane (perpendicular to the device) optical dipoles (such as associated with D$^0$). We label in-plane polarization as x, and out-of-plane polarization as z. We use this measurement scheme to resolve the dipole moment of transitions in **D3**. **b** Before straining the sample ($V_p = 0$), X$^-$ is present when selecting for x-polarized light, but D$^0$ appears when selecting for z-polarized light. **c** At $V_p = 10$ V, the red-shifted features (<1.6 eV) possess a much greater degree of z-polarization than the plateauing branch (~1.7 eV).

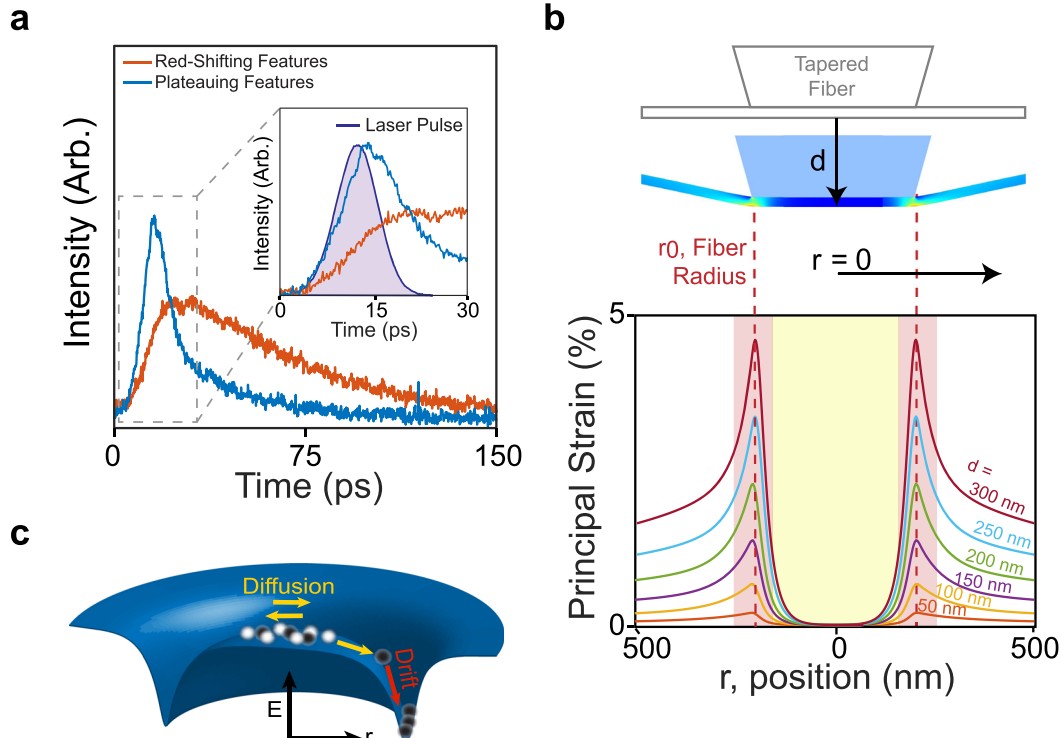

**Fig. 4 Dark and bright exciton navigation across a local strain landscape. a** Time-resolved PL from both the plateauing features and red-shifting features in device **D4** reveal lifetimes associated with each set of 5.4 ± 0.2 and 48.2 ± 0.6 ps, respectively. Inset: Close up of the first 30 ps of the time evolution, with the laser pulse shown. The red-shifted features have a ~10 ps delay relative to the plateauing features. **b** Finite element method modeling of strain in the hBN/WSe$_2$/hBN heterostructure as it is displaced a distance d by the fiber. The top part of the figure corresponds to $d = 300$ nm (strain of 5%). **c** Schematic of the strain-induced energy potential due to the fiber facet (dark blue surface). Bright (white spheres) and dark (black spheres) excitons navigate this potential through a combination of diffusion (yellow arrows) and drift (red arrow); only dark excitons have a long enough lifetime to reach the energetic minimum.

MoSe$_2$ monolayers may stem from the fact that the dark excitons are not the lowest energy excitonic state in that system, which provides an alternative explanation to that proposed in some theoretical works[32]. Finally, the ability to create energetic traps hundreds of meV deep via strain coupled with the long lifetime of dark excitons indicates a potential route for creating dark exciton condensates in a monolayer semiconductor[33,34].

## Methods

**Sample preparation.** Monolayer WSe$_2$ and MoSe$_2$ (HQ Graphene) and few-layer graphite flakes were exfoliated onto silicon substrates with a 285 nm silicon oxide layer. hBN flakes were exfoliated onto silicon substrates with a 90 nm silicon oxide layer. Monolayers of WSe$_2$ and MoSe$_2$ were identified by their contrast under an optical microscope and verified by their PL spectra. Both hBN/WSe$_2$/hBN and hBN/WSe$_2$/hBN/graphite heterostructures were fabricated by a dry transfer method. These heterostructures were then transferred to glass

substrates with pre-patterned pits formed by reactive ion etching (4 μm diameter, 0.5–2.0 μm deep) in order to be suspended. Next, electron-beam lithography was used to define contacts to the $WSe_2$ and few-layer graphite (where applicable) and deposited by thermal evaporation (5 nm chromium and 95 nm gold).

**Fiber preparation and fiber-based spectroscopy.** Commercial near-infrared single-mode optical fibers (Thorlabs S630-HP) are stripped of their buffer coating and cleaned before being submerged in hydrofluoric acid (HF) to generate their taper profile[35]. The taper tip is then deterministically etched by mounting the tapered fiber in a focused-ion beam (FIB) microscope. Standard ionic-clean (SC-2) is subsequently carried out to remove potential residuary ions (Ga+) from the FIB processing.

The fiber is placed using tweezers into a groove in a custom, copper sample mount on top of our piezoelectric nanopositioner stack (Attocube) and then clamped down with a copper plate to prevent it from sliding in the groove during operation. The tapered end of the fiber hangs out over the edge of the sample mount to facilitate bringing it in proximity with the sample. The heterostructures are also mounted on a stack of nanopositioners. The tapered fiber is navigated to the suspended $WSe_2$ heterostructure by monitoring its position with widefield imaging using two objectives. A 10× objective looking down the fiber axis allows us to position the fiber over the suspended region, while a 100× objective orthogonal to the fiber axis enables us to slowly bring the tip into contact with the sample (see also Supplementary Fig. 1, which shows a lamp where the 10× objective is during sample alignment). A 520 nm CW laser with 30–500 nW of power is coupled into the fiber to excite the sample and PL is collected along the same path, using a non-polarizing beamsplitter and long-pass filter at 590 nm to separate PL from reflected laser light.

**Polarization-selective optical spectroscopy.** With the sample and fiber mounted inside a Montana Instruments cryostat at 5 K, measurements were made through the side window using a home-built confocal microscope with a ×100, 0.9 NA objective (Olympus). The sample is excited through the fiber with 520 nm laser light and PL is collected confocally after a 715 nm long-pass filter. A half-wave plate rotates the polarization of the PL before passing through a linear polarizer.

**Time-resolved photoluminescence spectroscopy.** A fs-pulsed laser (Mira F-900) with a repetition rate of 79 MHz and with wavelength 800 nm is frequency-doubled to 400 nm and sent through the fiber to excite the sample. PL is collected through the same fiber and directed to a streak camera (Hamamatsu C5680 and Hamamatsu ORCA-IR) with a 2 ps resolution. A collection of tunable short-pass and long-pass filters are used to spectrally isolate certain PL features before sending them to the streak camera.

## Data availability
The data that support the plots within this paper and other findings of this study are available from the corresponding authors upon reasonable request.

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

## Acknowledgements
M.L. acknowledges ARO MURI grant no. W911NF1810432, STC Center for Integrated Quantum Materials, NSF grant no. DMR-1231319, and ONR MURI grant no. N00014-15-1-2761. H.P. acknowledges support from the DoD Vannevar Bush Faculty Fellowship (N00014-16-1-2825), NSF (PHY-1506284), NSF CUA (PHY-1125846), ARL (W911NF1520067), DOE (DE-SC0020115), and Samsung Electronics. B.U. acknowledges funding from ANR HiLight and the EUR grant NanoX n° ANR-17-EURE-0009 in the framework of the Programme des Investissements d'Avenir. K.W. and T.T. acknowledge support from the Elemental Strategy Initiative conducted by the MEXT,

Japan (Grant Number JPMXP0112101001) and JSPS KAKENHI (Grant Numbers 19H05790, 20H00354, and 21H05233). D.R. acknowledges support from the NSF GRFP and Ford Foundation fellowships. B.P. acknowledges financial support through a Horizon 2020 Marie Skłodowska-Curie Actions global fellowship (COHESiV, Project Number: 840968) from the European Commission. Device fabrication was performed at the Center for Nanoscale Systems (CNS), a member of the National Nanotechnology Coordinated Infrastructure Network (NNCI), which is supported by the National Science Foundation under NSF Grant No. 1541959.

## Author contributions

M.L. and H.P. conceived this project. R.J.G. and X.L. fabricated the samples and D.R. fabricated the tapered optical fibers. R.J.G., D.R., X.L., B.P., and B.U. designed and performed the measurements. S.B. and G.S. assisted with optical measurements. D.R. did the numerical simulations. R.J.G. and D.R. analyzed the data. T.T. and K.W. provided the hexagonal boron nitride samples. R.J.G., D.R., X.L., B.P., H.P., and M.L wrote the manuscript with extensive input from the other authors. M.L. and H.P. supervised the project. These authors contributed equally: R.J.G. and D.R.

## Competing interests

The authors declare no competing interests.
