## [Peer Review File · Nature Communications]

Reviewers' Comments:

Reviewer #1:

Remarks to the Author:

The manuscript builds up on previous reports of strain-induced quantum emission and demonstrates experimentally what was inferred from other measurements: That is, the dark exciton under local strain brightens under strain. The local nature of the strain profile gives the confinement of the excitons. They achieve this demonstration through a tapered optical fiber pressing on a suspended vdW layer. They display the spectral shift of optical transitions which qualitatively reproduces defect excitons or confined emission. Further, they claim, based on lifetime measurements, that the excitons are transported into the high-strain regions via transport.

Overall, I find all their arguments plausible, in fact none of it contradicts what was either reported before or inferred via previous measurements, including large g-factors and various strain-generation profiles). What is not present here is clear demonstration that these are localised emission peaks (Hanbury Brown Twiss type verification), which is necessary here to prove they are not simply seeing red-shifted emission, but they are also localised. This is what was reported in the Moon et al Nano Letters (2020), but the authors go beyond the claims and state the formation of localised emission. In my opinion, this is hard to tell apart in this work. Also, I did not see any reference to works based on AFM indentation, but perhaps I missed it. Another issue I have is the exciton diffusion claim based on change of lifetime response, but this can also come from multi-level mixing that is already taking place. So, it is indeed plausible, but not fully confirmed here. Finally, the authors should at least compare their results to the claims of Brooks et al., Phys. Rev. B 97, 195454 (2018), which is missing in their discussions or references.

In short, the idea is nice and it is good to see that they can apply enough strain to see significant change in the photoluminescence from this material. However, it is confirmation paper for the commonly accepted mechanism based on previous experimental results and they fall somewhat short of demonstrating unequivocally the diffusion and "navigation" which is part of the title. I suggest the work warrants publication, but it is not impactful enough to end remaining disputes on the origin of the localised emission.

Reviewer #2:

Remarks to the Author:

The results reported in the manuscript by Gelly, Renaud et al should be accepted in the current form for the following reasons:

- The optical setup, the fabrication process of the tapered fiber with nanometric tip, the 2D materials used and the process to fabricate the devices are explained in detail with data, images and schemes.
- The local, high-resolution measurement of the strain mapping was delicate and challenging at the experimental level, but the authors have been able to extract the spectra with the different excitons peaks and their lifetimes.
- The physical interpretation of the experimental results is consistent with the experimental data obtained in other works performed by other groups and also with theoretical modeling.
- The given bibliography is relevant and up to date.
- The obtained information about dark excitons in WSe₂ monolayer is relevant for the advancement of the straintronics field and it is also relevant to achieve more understanding in the physics of 2D transition metal dichalcogenides.

Reviewer #3:

Remarks to the Author:

Authors have come up with an interesting idea of using a tapered (and truncated with a top diameter of 480 nm) optical fiber for pushing a suspended membrane of layered materials (containing a WSe₂ monolayer) to create a dynamic-and-reversible strain in the WSe₂ monolayer.

COMSOL FEM simulation indeed support creation of such a strain. Authors also use the same fiber-tip for optically exciting the monolayer WSe₂ and for collecting both the PL and the transmittance signals.

The work presented by authors indeed shed some light to understand the origin of the observations of single photon emitters (SPEs) in the monolayer WSe₂.

Authors have provided an evidence of observing a bunch low-energy lines ONLY at large strain by collecting the PL signal from the side, and these lines are satisfying one of the criteria of a dark exciton that it would emit an out-of-plane polarized light.

Although, authors have taken an important step forward to understand these SPEs, the results and interpretations presented here, in my opinion, are not strong enough to guarantee a publication of this manuscript in the Nat. Comm. journal. Author may improve the contents of their manuscript by considering my following major and minor criticism/comments:

Major criticisms/comments:

(1) Authors have shown that the lifetimes of these emitters are in the timescale of 50 picoseconds, which is orders of magnitude smaller than the usually observed/measured lifetimes of SPEs in WSe₂ (usually in the range of 1 – 10 ns or so). Authors may consider performing some basic characterizations of single photon emission of an individual/localized emitter (; such as HBT measurement on a single line, g-factors of these lines, presence of zero-magnetic field splitting etc.) to gain some in-depth knowledge on these emitters.

(2) A few experimental work (e.g. Optica 10, 507, 2015; Nano Lett. 15, 7567, 2015; Adv. Matter 28, 7101, 2016), and a recent theoretical work (, Ref. 12 of this manuscript) say that the observation of SPEs in WSe₂ is a strain-induced phenomenon. The crystal defects, that are present in the strained region, are activated because of a strain-induced change in the bandgap and funnelling of (dark) excitons. Another hand waving explanation is that local change in bandgap is significant enough to form a quantum dot like confinement region for trapping the excitons. Although authors are mentioning that their results is supporting the theoretical results of Ref. 12, they do not mention anything about crystal defects and their activation due to a large strain.

(3) Authors claim that dark exciton diffuse/drift from fiber-facet region to fiber-circumference region, and therefore, they see a decent rise time. As the fiber-tip is illuminating both the facet and the circumference regions, the excitons (bright/dark) are also created right in the circumference region, the claim of a rise time is not justified.

Minor comments:

(1) Authors may shed some light on broadening of XA:1s transmittance dip at a higher strain.

(2) It is not clear in the text that how authors attach a tapered fiber with a piezo positioner.

List of revisions

- Replaced throughout “excitons” and “dark excitons” with “free excitons” and “free dark excitons” where appropriate to clarify that we are not looking at quantum confined states
- Highlighted the differences between our work and that of Moon *et al.* in paragraph 3 (page 3)
- Clarified language in paragraph 3 (page 3) to avoid confusion over free vs. defect-bound excitons
- Added five references: one on exciton transport (Ref. 15), two on AFM indentation (Refs. 19, 21), one on carrier dynamics (Ref. 34), and one theoretical proposition (Ref. 37)
- Added Supplementary Figure 7 with numerical solutions to the drift diffusion equation that support delay in rise time due to transport
- Added a reference to Supplementary Figure 7 in the main text in Paragraph 11 (page 8)
- Added a discussion sentence in paragraph 13 (page 9) regarding the role of defects
- Added a discussion of why the $X_{A:1s}$ state broadens with strain in paragraph 5 (page 5)
- Updated the methods section with additional detail on how we mount the fiber on the nanopositioner and how we bring the fiber close to the sample for contact

Point-by-point response to the referees

Reviewer #1 (Remarks to the Author):

The manuscript builds up on previous reports of strain-induced quantum emission and demonstrates experimentally what was inferred from other measurements: That is, the dark exciton under local strain brightens under strain. The local nature of the strain profile gives the confinement of the excitons. They achieve this demonstration through a tapered optical fiber pressing on a suspended vdW layer.

We thank the reviewer for his/her careful reading of the manuscript and for summarizing the salient points. We would like to emphasize that in addition to providing an important experimental verification of previous predictions and inferences, we present a systematic study of the same region of 2-D material with and without strain introduced. This allows us to **study and understand the origin of the extremely red shifted emission from the same sample under high strain.**

They display the spectral shift of optical transitions which qualitatively reproduces defect excitons or confined emission.

While the emission is indeed produced from excitons trapped by the strain field, we do not believe that our work is reproducing defect excitons or quantum confined emission.

Specifically, we do not observe sharp linewidths ($< 1\text{meV}$), and when we measured the second-order correlation function via a Hanbury Brown Twiss (HBT) measurement at high strain, we did not observe $g^2(0) < 1$. Therefore, we believe we are looking exclusively at free excitons. This is consistent with the size of the high-strain region (highlighted red in Fig. 4b) being tens to hundreds of nanometers, compared to an excitonic Bohr radius of 1 nm. Moreover, the gate dependence at high strain (Figure S4) is similar in character to that of the free excitons at low strain, distinct from the stepped structure of single photon sources in WSe₂ (Ref. 6).

Further, they claim, based on lifetime measurements, that the excitons are transported into the high-strain regions via transport.

Overall, I find all their arguments plausible, in fact none of it contradicts what was either reported before or inferred via previous measurements, including large g-factors and various strain-generation profiles). What is not present here is clear demonstration that these are localised emission peaks (hanbury brown twiss type verification), which is necessary here to prove they are not simply seeing red-shifted emission, but they are also localised.

We regret that our original manuscript failed to convey our main findings clearly: we are not claiming the quantum confinement or quantum localization of excitons in this work. Rather, we are focusing on how dark excitons, which are a necessary ingredient for the formation of quantum emitters, travel through a local strain landscape. **We have clarified this point by replacing “excitons” and “dark excitons” with “free excitons” and “free dark excitons” where appropriate throughout the manuscript.**

This is what was reported in the Moon et al Nano Letters (2020), [...]

We believe that our work goes beyond the nice work of Moon et al. in several ways. The three most essential differences are:

(i) We are able to distinguish between the excitonic species that govern emission due to the high optical quality of our hBN encapsulated samples and the presence of a gate for charge tuning studies. Very importantly, we can distinguish nominally dark from bright excitons through polarization resolved studies.

(ii) Our newly developed technique enabled us to perform both photoluminescence and transmission studies on strained states in the same sample on the same spot and compare the results with those from the zero-strain regime.

(iii) We observe the considerably larger shifts to exciton energies (up to 390 meV, compared to ~20 meV in their work), allowing us to probe a very different strain regime.

We now more clearly contextualize the work of Moon et al. and highlight its difference from our work by the addition of the text:

(Paragraph 3, page 3) *Important advancements towards dynamically applied strains in a cryogenic environment have so far remained in the low strain regime ($< 0.5\%$)².*

[...] but the authors go beyond the claims and state the formation of localised emission.

We have clarified our language to distinguish between localized emission and local emission – what we observe are not quantum-confined states, but rather emission from states from a spatial region that is small compared to the sample size, but large compared to the exciton Bohr radius. **To clarify, we made the following change to the manuscript:**

(Paragraph 3, page 3) *Here, we systematically address the role of locally applied strain on free (as opposed to defect-bound) excitonic species in WSe₂ monolayers by dynamically generating sub-micron scale (~100 nm) strain at cryogenic temperatures.*

In my opinion, this is hard to tell apart in this work. Also, I did not see any reference to works based on AFM indentation, but perhaps I missed it.

We have added the following references regarding AFM indentation to our introductory paragraphs, in addition to the currently cited paper by Moon et al. that also utilizes AFM indentation.

[19] Rosenberger, M. R. *et al.* Quantum Calligraphy: Writing Single-Photon Emitters in a Two-Dimensional Materials Platform. *ACS Nano* **13**, 904-912, doi:10.1021/acsnano.8b08730 (2019).

[21] De Palma, A. C. *et al.* Strain-dependent luminescence and piezoelectricity in monolayer transition metal dichalcogenides. *Journal of Vacuum Science & Technology B* **38**, doi:10.1116/6.0000251 (2020).

Another issue I have is the exciton diffusion claim based on change of lifetime response, but this can also come from multi-level mixing that is already taking place. So, it is indeed plausible, but not fully confirmed here.

We thank the reviewer for the opportunity to clarify our time-dependent data. As for the decay time changing via multi-level mixing, we point out that it is the change in rise time delay, not decay time, which we present as evidence of exciton diffusion. We do agree with the reviewer that the decay time will be convoluted by multi-level mixing.

While the rise time may also be affected by multi-level mixing, the work by Bertoni *et al.* (now Ref. 34) looked at the delay in populating states across k space and found that dynamics occurred in < 100 fs. The process we are observing is slower. Therefore, we believe the delay in rise time is best described by the exciton transport, especially in the context of the other pieces of supporting evidence in our manuscript.

We would also point to our study of MoSe₂, which we discuss in Paragraph 12 (page 8), a system in which the ground exciton state is the short-lived bright exciton. Here, we find that there is no red-shifting branch. When the ground state is so short-lived, there is no appreciable transport prior to decay, and thus we do not see significant population at the boundary.

We have added a sentence in our discussion of the time-resolved PL data to address this good point by the reviewer, which includes a reference to the aforementioned work by Bertoni *et al.*. It reads:

(Paragraph 11, page 7) Other processes, such as the scattering from other states across reciprocal space³⁴, are too fast to explain this long delay.

We have also added a section to the Supplementary Information that expands upon our argument with numerical simulation results. We believe that this additional evidence allows us to argue our case more effectively. The new content is reproduced below:

(Paragraph 11, page 8) Numerical solutions to the drift-diffusion equation also capture this behavior qualitatively (Supplementary Figure 7).

(Supplementary Information)

Figure S8 – Numerical Modeling of the Drift-Diffusion Equation

To verify our model of the dynamics in the system, we numerically solve a drift-diffusion partial differential equation for the exciton population n :

$$\frac{\partial n(x, t)}{\partial t} = D \nabla^2 n(x, t) - \mu \nabla \cdot (\mathbf{F}(x) n(x, t)) + I(x, t) - \frac{n(x, t)}{\tau}$$

The first term on the right-hand side is the diffusive term, with diffusivity D . The second is the drift term with mobility μ under the influence of a force field \mathbf{F} , here derived from our strain potential $\mathbf{F} = -\nabla U = -\nabla(C\epsilon)$ for ϵ the strain and C the coefficient giving bandgap shift per strain. The third term is the laser intensity which populates the exciton states. The fourth is exciton decay during their lifetime τ . We take $D = 1 \text{ cm}^2/\text{s}$, calculate the mobility using the Einstein relation at temperature $T = 23 \text{ K}$, and take the exciton lifetimes from our streak camera measurements. (a) Depicts the function $U(x)$ for a well depth of -0.1 eV (blue curve) as well as the laser profile which sets the spatial profile of $I(x, t)$. The temporal behavior of $I(x, t)$ is a Gaussian with 2 ps width. (b) The time evolution of the exciton population at both the fiber center (center) and the well minimum (edge) is plotted for bright (X^0) and dark (D^0) excitons, with the only difference between the two being a lifetime of $\tau = 5 \text{ ps}$ (bright) or $\tau = 48 \text{ ps}$ (dark), matching the data in Fig. 4. This model qualitatively captures the delay in population at the fiber edge observed in the time-resolved PL measurements.

References

- 1 Cordovilla Leon, D. F., Li, Z., Jang, S. W., Cheng, C.-H. & Deotare, P. B. Exciton transport in strained monolayer WSe₂. *Applied Physics Letters* **113**, doi:10.1063/1.5063263 (2018).

Finally, the authors should at least compare their results to the claims of Brooks et al., Phys. Rev. B 97, 195454 (2018), which is missing in their discussions or references.

We thank the reviewer for pointing out this missing reference. We fully agree that the discussion in Brooks *et al.* contains the general concept we discuss in this work: strain engineering. The strain profiles generated by nano-pillars discussed by Brooks *et al.* is some ways similar to the non-uniform strain profile we discuss for the fiber tip. Our work shows that in addition to strain engineering of the bandgap, the existence of long-lived dark states has an important impact on the PL emission under strain. **We now include a reference to the work by Brooks et al. (Ref. 37) in our conclusion and discuss how our results provide an alternative reason for the scarcity of emitters in MoSe₂ compared to the band-structure-related claims in that paper.** The new text reads (changes in bold):

(Paragraph 13, page 9) Our results also suggest that the scarcity of strain-induced single-photon sources in MoSe₂ monolayers may stem from the fact that the dark excitons are not the lowest energy excitonic state in that system. **We note that this explanation provides an alternative explanation to that proposed in some theoretical works³⁷.**

In short, the idea is nice and it is good to see that they can apply enough strain to see significant change in the photoluminescence from this material.

We thank the reviewer for recognizing the significant tuning by over 350 meV of the photoluminescence in our work.

However, it is confirmation paper for the commonly accepted mechanism based on previous experimental results and they fall somewhat short of demonstration unequivocally the diffusion and "navigation" which is part of the title. I suggest the work warrants publication, but it is not impactful enough to end remaining disputes on the origin of the localised emission.

We thank the reviewer for finding our work worthy of publication. We would like to point out that while exciton navigation to defects has been a commonly inferred mechanism in prior work, the role of dark excitons has previously been unclear or at least underemphasized. Our work addresses this and clearly delineates the roles that the bright and dark excitons plays. In addition, our work adds the new and important ingredient that is strain tuning, which allows the direct comparison of the zero strain emission with highly red shifted transitions when high strain is applied. Identifying the role of the dark exciton required the employment of polarization- and time-resolved spectroscopic techniques and multiple material systems (WSe₂ vs. MoSe₂).

Reviewer #2 (Remarks to the Author):

The results reported in the manuscript by Gelly, Renaud et al should be accepted in the current form for the following reasons:

-The optical setup, the fabrication process of the tapered fiber with nanometric tip, the 2D materials used and the process to fabricate the devices are explained in detail with data, images and schemes.

-The local, high-resolution measurement of the strain mapping was delicate and challenging at the experimental level, but the authors have been able to extract the spectra with the different excitons peaks and their lifetimes.

-The physical interpretation of the experimental results is consistent with the experimental data obtained in other works performed by other groups and also with theoretical modeling.

-The given bibliography is relevant and up to date.

-The obtained information about dark excitons in WSe₂ monolayer is relevant for the advancement of the straintronics field and it is also relevant to achieve more understanding in the physics of 2D transition metal dichalcogenides.

We thank the reviewer for their careful reading of the manuscript, as well as for their positive comments and enthusiastic reception of our work. We hope that the changes we have made in response to the other reviewers will strengthen the reviewer's support for publishing this work.

Reviewer #3 (Remarks to the Author):

Authors have come up with an interesting idea of using a tapered (and truncated with a top diameter of 480 nm) optical fiber for pushing a suspended membrane of layered materials (containing a WSe₂ monolayer) to create a dynamic-and-reversible strain in the WSe₂ monolayer. COMSOL FEM simulation indeed support creation of such a strain. Authors also use the same fiber-tip for optically exciting the monolayer WSe₂ and for collecting both the PL and the transmittance signals.

The work presented by authors indeed shed some light to understand the origin of the observations of single photon emitters (SPEs) in the monolayer WSe₂.

Authors have provided an evidence of observing a bunch low-energy lines ONLY at large strain by collecting the PL signal from the side, and these lines are satisfying one of the criteria of a dark exciton that it would emit an out-of-plane polarized light.

We thank the reviewer for their careful reading of the manuscript and recognizing the appeal of our fiber-based approach to introduce strain and collect emission from the same region.

Although, authors have taken an important step forward to understand these SPEs, the results and interpretations presented here, in my opinion, are not strong enough to guarantee a publication of this manuscript in the Nat. Comm. journal. Author may improve the contents of their manuscript by considering my following major and minor criticism/comments:

We thank the reviewer for the opportunity to improve our work and will address the comments one-by-one below.

Major criticisms/comments:

(1) Authors have shown that the lifetimes of these emitters are in the timescale of 50 picoseconds, which is orders of magnitude smaller than the usually observed/measured lifetimes of SPEs in WSe₂ (usually in the range of 1 – 10 ns or so). Authors may consider performing some basic characterizations of single photon emission of an individual/localized emitter (; such as HBT measurement on a single line, g-factors of these lines, presence of zero-magnetic field splitting etc.) to gain some in-depth knowledge on these emitters.

As mentioned in our response to Reviewer 1, we now realize that in the original manuscript we should have been clearer regarding the fact that our system *does not* operate in quantum confined/localized regime. Our strain landscape's length scale (~100 nm) is too large compared to the exciton Bohr radius (~1 nm) for quantum confinement effects to be relevant. **Indeed, we are not claiming to observe single photon emitters (SPE) formed via quantum confinement or quantum localization of excitons in this work. Instead, we have measured the properties of the free excitons that are a precursor for quantum emitters (Ref. 12).** In particular, we investigate the role that dark excitons play in the emission process. Consequently, we expect the lifetime of these free dark excitons to be ~50-100 ps (Ref. 32). This lifetime is consistent with our current measurement of ~50 ps, and much shorter than that of quantum emitters, which the reviewer correctly identifies as being 1-10 ns. Moreover, because we are looking at free excitons

and not quantum emitters, HBT measurements do not show a dip at $g^{(2)}(0)$, nor do the spectral features exhibit a zero-field splitting. We have confirmed this experimentally: we measured HBT from several of the lines in the high-strain regime and verified that none had $g^{(2)}(0) < 1$. This, together with the lifetime measurements, confirms that we are not dealing with quantum confined excitons. **We have clarified this in the manuscript by saying:**

(Paragraph 8, Page 6) Moreover, because the lifetimes are on the picosecond scale, we know we are looking at free excitons, as opposed to single-photon emitter states with lifetimes greater than 1 ns.

(2) A few experimental work (e.g. Optica 10, 507, 2015; Nano Lett. 15, 7567, 2015; Adv. Matter 28, 7101, 2016), and a recent theoretical work (, Ref. 12 of this manuscript) say that the observation of SPEs in WSe2 is a strain-induced phenomenon. The crystal defects, that are present in the strained region, are activated because of a strain-induced change in the bandgap and funnelling of (dark) excitons. Another hand waving explanation is that local change in bandgap is significant enough to form a quantum dot like confinement region for trapping the excitons. Although authors are mentioning that their results is supporting the theoretical results of Ref. 12, they do not mention anything about crystal defects and their activation due to a large strain.

We respond to this point with similar remarks to the previous point: because these are free excitons, we do not consider their binding to defects in this work. Additionally, quantum dot-like confinement does not seem to be relevant for the strain confinement scales (~ 100 nm) in our experiment. **We have clarified that, in order to go from the free exciton picture we are presenting to a quantum emitter picture, binding to defects is necessary. This change is reflected in:**

(Paragraph 13, page 9) While it has been understood that these single-photon sources likely originate from the binding of excitons to defects, it has not been clear how the excitons reach the defects efficiently, nor has the essential role of the dark exciton been previously identified experimentally.

(3) Authors claim that dark exciton diffuse/drift from fiber-facet region to fiber-circumference region, and therefore, they see a decent rise time. As the fiber-tip is illuminating both the facet and the circumference regions, the excitons (bright/dark) are also created right in the circumference region, the claim of a rise time is not justified.

We have not described our scenario very clearly, and we thank the referee for pointing out these shortcomings. In order to arrive at the conclusion that we observe spatial diffusion of dark excitons, we start our arguments as follows :

- As we demonstrate in Fig. 1b, the excitation profile is heavily weighted toward the center of the fiber and the initial population at the boundary is small.
- Strain at the center of the fiber is lower than on the fiber boundary (high strain = lower optical transition energy)

- So the starting point is a high exciton population in the low strain region and a small exciton population at the high strain region. As a result, excitons will diffuse towards the high strain (low transition energy) region
- Only after diffusing or drifting to the edge (which takes some time) is the population there maximized. Finite-difference time-domain (FDTD) simulations of the drift-diffusion equation also supports this fact.
- We have added such simulations as a new section in the supplementary information as additional evidence. This model incorporates the initial laser profile explicitly. We also reproduce this new content below:

(Paragraph 11, page 8) Numerical solutions to the drift-diffusion equation also capture this behavior qualitatively (Supplementary Figure 7).

(Supplementary Information)

Figure S8 – Numerical Modeling of the Drift-Diffusion Equation

To verify our model of the dynamics in the system, we numerically solve a drift-diffusion partial differential equation for the exciton population n :

$$\frac{\partial n(x, t)}{\partial t} = D\nabla^2 n(x, t) - \mu\nabla \cdot (\mathbf{F}(x)n(x, t)) + I(x, t) - \frac{n(x, t)}{\tau}$$

The first term on the right-hand side is the diffusive term, with diffusivity D . The second is the drift term with mobility μ under the influence of a force field \mathbf{F} , here derived from our strain potential $\mathbf{F} = -\nabla U = -\nabla(C\varepsilon)$ for ε the strain and C the coefficient giving bandgap shift per strain. The third term is the pulsed laser intensity which populates the exciton states. The fourth is exciton decay during their lifetime τ . We take $D = 1 \text{ cm}^2/\text{s}$, calculate the mobility using the Einstein relation at temperature $T = 23 \text{ K}$, and take the exciton lifetimes from our streak camera measurements. (a) Depicts the function $U(x)$ for a well depth of -0.1 eV (blue curve) as well as the laser profile which sets the spatial profile of $I(x, t)$. The temporal behavior of $I(x, t)$ is a Gaussian with 2 ps width. (b) The time evolution of the exciton population at both the fiber center (center) and the well minimum (edge) is plotted for bright (X^0) and dark (D^0) excitons, with the only difference between the two being a lifetime of $\tau = 8 \text{ ps}$ (bright) or $\tau = 53 \text{ ps}$ (dark), matching the data in Fig. 4. This model qualitatively captures the delay in population at the fiber edge observed in the time-resolved PL measurements.

References

- 1 Cordovilla Leon, D. F., Li, Z., Jang, S. W., Cheng, C.-H. & Deotare, P. B. Exciton transport in strained monolayer WSe₂. *Applied Physics Letters* **113**, doi:10.1063/1.5063263 (2018).

Minor comments:

(1) *Authors may shed some light on broadening of X_{A:1s} transmittance dip at a higher strain.*

We thank the reviewer for this question. Two possible explanations include:

1. Inhomogeneous broadening due to a non-uniform strain across the fiber facet
2. An increase in non-radiative decay due to coupling of different valleys by strain

We have added a comment to address this in the main text:

(Paragraph 5, page 5) We also observe some broadening in the X_{A:1s} state that may be due to inhomogeneous broadening stemming from non-uniformity in the strain imposed by the fiber facet. The broadening may also originate from non-radiative broadening due to coupling of different valleys by strain.

(2) *It is not clear in the text that how authors attach a tapered fiber with a piezo positioner.*

We have modified the methods section to describe how we contact the tapered fiber to the suspended sample with the piezo positioner:

(Methods, page 17) The tapered fiber is navigated to the suspended WSe₂ heterostructure by monitoring its position with widefield imaging using two objectives. A 10x objective looking down the fiber axis allows us to position the fiber over the suspended region, while a 100x objective orthogonal to the fiber axis enables us to slowly bring the tip into contact with the

sample (see also Supplementary Figure S1, which shows a lamp where the 10x objective is during sample alignment).

We have also modified the methods section to describe the attachment of the tapered fiber to the piezo positioner. It reads:

(Methods, page 16) The fiber is placed using tweezers into a groove in a custom, copper sample mount on top of our nanopositioner stack and then clamped down with a copper plate to prevent it from sliding in the groove during operation. The tapered end of the fiber hangs out over the edge of sample mount to facilitate bringing it in proximity with the sample.

Reviewers' Comments:

Reviewer #1:

Remarks to the Author:

The authors have taken on my suggestions and responded satisfactorily to my open queries. I am particularly happy about the clarifications across the paper about confined versus trapped excitons. The diffusion model also seems sound and explains the lifetime observations well. In its revised form, I am happy to recommend that this manuscript is of sufficient quality and rigour to be published in Nature Communications.

Reviewer #3:

Remarks to the Author:

Authors have implemented/addressed the earlier corrections/criticisms/comments. Now the manuscript is providing a detailed interrogations of free dark excitons in the ML WSe₂. It shall be published without any further revisions.

Point-by-point response to the referees

Reviewer #1 (Remarks to the Author):

The authors have taken on my suggestions and responded satisfactorily to my open queries. I am particularly happy about the clarifications across the paper about confined versus trapped excitons. The diffusion model also seems sound and explains the lifetime observations well. In its revised form, I am happy to recommend that this manuscript is of sufficient quality and rigour to be published in Nature Communications.

We thank the reviewer for the encouraging comments in support of publication.

Reviewer #3 (Remarks to the Author):

Authors have implemented/addressed the earlier corrections/criticisms/comments. Now the manuscript is providing a detailed interrogations of free dark excitons in the ML WSe₂. It shall be published without any further revisions.

We thank the reviewer for the encouraging comments in support of publication.